# Learning Generalizable World Models via Discovering Superposed Causal Relationships

## Abstract

In reinforcement learning, a generalizable world model to mimic the environment is crucial for the assessment of various policy values in downstream tasks such as offline policy optimization and off-policy evaluation. Recently, studies have shown that learning a world model with sparse connections identified by causal discovery techniques can improve generalizability. So far, these studies focus on discovering a single and global causal structure. In this paper, we discuss a more practical setting in which the agent is deployed in an environment mixed with different causal mechanisms, called superposed causal relationships in this article. In this case, global causal discovery techniques will derive a degraded dense causal relationship, which will fail to improve the generalizability of the learned model. To solve the problem, we propose **S**uperposed c**A**usal **M**odel (SAM) learning. SAM learning is an end-to-end framework that learns a transformer-based model which can recognize the causal relationships that the agent is encountering on the fly and then adapts its predictions. The experiments are conducted in two simulated environments, where SAM shows powerful identify abilities in environments with superposed causal relationships. Both the dynamics model and the policies learned by the SAM generalize well to unseen states.

## 1 Introduction

Learning an accurate environment model that approximates state transitions is crucial in fields such as offline reinforcement learning (offline RL) (Levine et al., 2020b). By utilizing world models, costly real-world trial-and-error processes can be avoided. The primary role of these models is to unbiasedly answer counterfactual queries; that is, given certain states, they can correctly predict *what might happen if* we were to execute actions unseen in the training data. Previous studies have shown that learning a causal world model—which identifies causal relationships and employs sparse connections for environment model learning—can improve generalization ability (Wang et al., 2022).

In these studies, sparse connection patterns are identified using causal structure discovery techniques (Ding et al., 2022). These patterns skip connections between the current state-action pair and the next state when no causal relationship exists, enhancing model efficiency. Unfortunately, most existing methods assume that a single causal model governs the entire dataset. However, in many real-world decision-making tasks, data exhibit multimodal characteristics and substantial heterogeneity. For instance, in dynamic resource allocation tasks, the causal relationships between actions and outcomes can differ across various contexts or states within the same environment, reflecting an underlying multimodal causal structure. Similarly, the causal relationships between driving strategies and environmental feedback can vary under different traffic conditions (such as rush hour vs. non-rush hour, city streets vs. highways), highlighting the multimodal nature of causal structures within the same environment.

In this paper, we aim to advance causal world model learning by transitioning from *single* causal world model learning to *superposed* causal world model learning, indicating that the environments we encounter encompass multiple causal relationships. Currently, the challenge of identifying mixtures of causal graphs from data has not been extensively explored in the literature. Most recent works, such as (Varambally et al., 2024), have addressed the challenge of inferring causal models from mixture distributions. However, these approaches do not tackle sequential decision-making tasks.

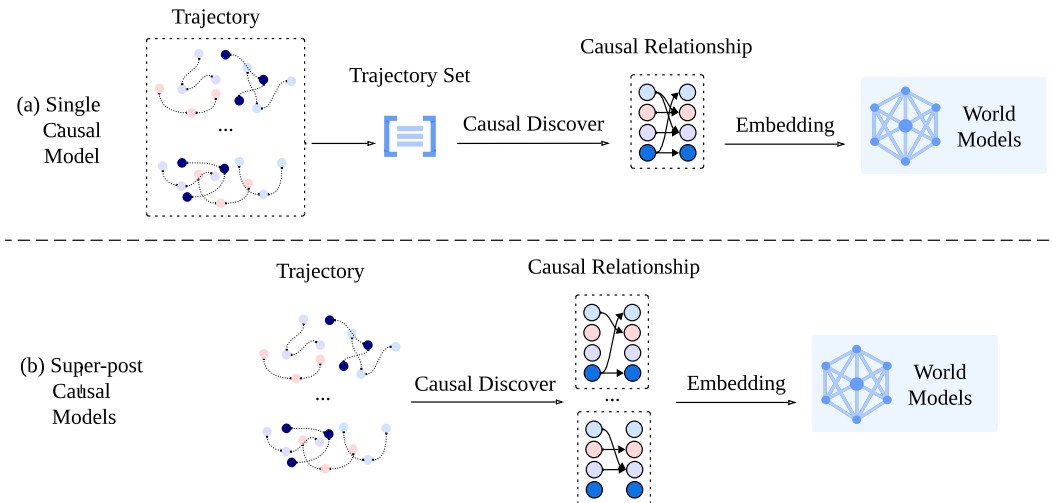

Figure 1: (a) Previous problems focus on causal discovery on trajectory set, where all the trajectories share a common dense causal graph. (b) We focus on the causal discovery by each trajectory, where each trajectory has different sparse causal graphs. In this setting, using global causal graph with finding a over-dense graph.

Specifically, we consider environments for decision-making that include an unknown number of causal relationships. Each time an episode is reset, a new causal relationship is generated, affecting the transition process and the optimal action (see Figure 1 for detailed comparison). To this end, we propose a causal world model learning algorithm, **S**uperposed c**A**usal **M**odel (SAM) learning. SAM is a general end-to-end world model learning algorithm that identifies the causal relationships of the current episode on the fly and predicts transitions based on the identified causal relationships. SAM is built upon the Transformer architecture to infer causal relationships from past interaction trajectories, with a unified training objective based on the evidence lower bound of the trajectory generation likelihood in our setting. Thanks to the minimalist design of the implementation, SAM is generally effective across different tasks and environments. We verify this through extensive experiments.

## 2 RELATED WORK

### 2.1 CAUSAL WORLD MODEL IN REINFORCEMENT LEARNING

Model-based RL improves policy learning by constructing environment models, including dynamics and reward predictors, through maximizing the likelihood of collected trajectories (Moerland et al., 2023). In online RL, these models enhance sample efficiency by aiding planning (Buesing et al., 2018; Hafner et al., 2019; Argenson & Dulac-Arnold, 2021), generating synthetic data (Janner et al., 2019; Hafner et al., 2020; Kaiser et al., 2020), or refining Q-value estimates (Feinberg et al., 2018; Amos et al., 2021). For offline RL (Levine et al., 2020a), the learned model facilitates effective offline policy evaluation (Argenson & Dulac-Arnold, 2021) and policy optimization by generating counterfactual trajectories (Yu et al., 2020; Kidambi et al., 2020).

Integrating causality into reinforcement learning (RL) has attracted substantial interest due to its benefits in generalization and transferability, particularly in representation learning (Huang et al., 2022; Liu et al., 2023), policy learning (Buesing et al., 2019; Mozifian et al., 2020), and dynamics learning (Wang et al., 2022; Ding et al., 2022; Hwang et al., 2024; Zhu et al., 2022). Within the realm of causal dynamics learning, Wang et al. (2022) explicitly learn causal dependencies by regulating the number of variables used to predict each state variable. Ding et al. (2022) enhance Goal-Conditioned RL with a causal graph to create generalizable models and interpretable policies. Hwang et al. (2024) propose a dynamics model that infers fine-grained causal structures, using a discrete latent variable to segment the state-action space to different fine-grained graph for model prediction. Zhu et al. (2022) theoretically demonstrate that causal world-models outperform non-causal models in offline RL by integrating causal structures into generalization error bounds. Unlike

these approaches that discover a single, global causal structure, this paper addresses a more practical scenario where the agent operates in an environment with mixed causal mechanisms, termed superposed causal relationships. To the best of our knowledge, we are the first to address superposed causal relationships in model-based RL, demonstrating that this approach not only identifies ground-truth causal structures but also outperforms baselines in generalization.

## 2.2 MULTIPLE CAUSAL STRUCTURED GRAPH DISCOVERY

Recently, researchers have begun exploring causal discovery in scenarios where causal relationships between variables vary across domains. Non-deep learning methods, such as SSCM (Huang et al., 2019), model both variability and consistency in causal relationships, leveraging commonalities to achieve statistically reliable estimates while preserving individual causal characteristics. Markham et al. (2022) proposed a kernel function based on distance covariance to measure causal structure similarity, identifying homogeneous subgroups through clustering. However, these methods often require manual design of scoring functions and can be computationally intensive, limiting scalability to large datasets. On the other hand, neural network-based approaches (Löwe et al., 2022; Lorch et al., 2022; Varambally et al., 2024) treat causal discovery as a black-box problem, assuming a distribution in the data and using variational inference to uncover domain-specific inductive biases. Our method is inspired by them but differs in that we consider environments for decision-making that include an unknown number of causal relationships. Each time an episode is reset, a new causal relationship is generated, affecting the transition process and the optimal action. However, these methods often require manual scoring function design and can be computationally intensive, limiting scalability to large datasets. In contrast, neural network-based approaches(Löwe et al., 2022; Lorch et al., 2022; Varambally et al., 2024) treat causal discovery as a black-box problem, assuming data distribution and using variational inference to uncover domain-specific biases. Our method is inspired by these but differs by focusing on decision-making environments with an unknown number of causal relationships. Each episode reset introduces a new causal relationship, affecting transitions and optimal actions, creating a novel RL setting distinct from previous multiple causal structured graph discovery literature.

## 3 PRELIMINARIES

### 3.1 MARKOV DECISION PROCESSES

We consider the environment within the framework of Markov Decision Process (MDP) setting with full observation defined as $\mathcal{M} = (\mathcal{S}, \mathcal{A}, \mathcal{P})$, where $\mathcal{S}$ and $\mathcal{A}$ represent the state and action space, $\mathcal{P}$ represents transition function. Following Seitzer et al. (2021), we assumed that the state space $\mathcal{S}$ can be factorized into $d^s$ disjoint components $\{S^1 \times \cdots \times S^d\}$, denoted as $\{\mathcal{S}^i\}_{i=1}^d$. similarly, $\mathcal{A}$ can be factorized into $d^a$ disjoint components.

### 3.2 STRUCTURED CAUSAL MODELS

Considering $d$ random variables $X = (X^1, \ldots, X^d)$ and a Directed Acyclic Graph (DAG) $\mathcal{G} = (V, E)$, where each node $i \in V = \{1, \ldots, d\}$ corresponds to a random variable $X^i$, a Structured Causal Model (SCM)(Pearl (2009)) is defined by a probability distribution $p_X$ over $X$ and the graph $\mathcal{G}$. In this context, each directed edge $(i \to j) \in E$ signifies that $X^i$ is a direct cause of $X^j$. This causal relationship implies that $X^j$ is conditionally dependent on $X^i$ during the data generation process. Denoting the set of parent nodes of node $i$ in the graph $\mathcal{G}$ as $Pa_i$, the joint distribution of the random variables can be formulated as:

$$p_X(x^1, \ldots, x^d) = \prod_{i=1}^d p_{X^i}(x^i | Pa_i), \tag{1}$$

# 4 SUPERPOSED CAUSAL ENVIRONMENT MODEL LEARNING

In this section, we introduce propose **S**uperposed c**A**usal **M**odel (SAM) learning. SAM identifies corresponding causal relationships from trajectories and utilizes them to learn the transition model in an end-to-end way. We begin by formalizing the problem, then we propose our approach to learn the superposed causal structure that utilizes individual trajectory information as well as group information.

## 4.1 PROBLEM FORMULATION

Different from standard SCMs discovery (Pearl, 2009), the Markov Decision Process under causal relationships can be characterized by one-step transition dynamics $f$ utilizing a causal mask $\mathcal{G}$ of dimensions $(d^s + d^a) \times d^s$. The nodes of this mask correspond to the set of random variables $\mathcal{V} = \{S_t^1, \ldots, S_t^{d^s}, A_t^1, \ldots, A_t^{d^a}, S_{t+1}^1, \ldots, S_{t+1}^{d^s}\}$. The subsequent state $s_{t+1}$ follows $p(s_{t+1}|s_t, a_t) = \prod_{i=1}^d p(s_{t+1}^i|s_{t+1}, a_t)$. The causal graph $\mathcal{G}$ here are mask matrices to represent the causal relationship between the two consecutive timesteps, each node represents a factor of the state, and each edge encapsulates a causal relationship. A factor of state $s_{t+1}^i$ follows $p(s_{t+1}^i|s_{t+1}, a_t) = p(s_{t+1}^i|Pa(s^i)_t, \mathcal{G})$, where $Pa$ represents the set of parent nodes in $\mathcal{G}$.

We consider the problem under the framework of Offline Reinforcement Learning (RL) using a superposed causal dataset $\mathcal{D} = \{\{\tau_i\}_{i=1}^N\}$. This superposed causal dataset is collected from $C$ environments that share the same decomposition but exhibit different causal relationships $\mathcal{G}_i$ for each environment $i$; that is, $\mathcal{D} = \{\mathcal{D}^{\mathcal{G}_i}\}_{i=1}^C$. We note that $C$ is an unknown in prior. Each trajectory $\tau_i$ is collected from one environment that every step is consistent with causal relationship.

Superposed causal relationship of dataset complicates the identification of distinct causal masks through time-series trajectories, which will introduce wrong or spurious causal relationship when learning dynamics. Thus, we aim to learn dynamics model with decomposed causal graph, ultimately enhancing the model's generalizability.

## 4.2 END-TO-END SUPERPOSED CAUSAL WORLD MODEL LEARNING

In previous RL research, dynamics with causal relationships typically learn causal masks using search-based methods. For example, CDL (Wang et al., 2022) utilizes conditional mutual information and considers a value higher than a threshold as indicative of a causal relationship, while GRADER (Ding et al., 2022) conducts statistical independence tests. Since search-based methods have high computational cost, cannot scale to larger environments, and require carefully designed search processes tailored to the task, this paper aims to propose a method that is scalable in terms of data volume. Therefore, unlike previous work, we employ a score-based method to discover causal masks from offline datasets, which models the entire causal graph discovery as a differentiable optimization objective, enabling end-to-end graph discovery and prediction. We derive the optimization objective of the superposed causal world model using a similar approach to Varambally et al. (2024):

$$\min_{\phi,\theta} -\mathbb{E}_{\tau \sim \mathcal{D}} \left[ \mathbb{E}_{q_\phi(\mathcal{G})} \left[ \log p_\theta(\tau \mid \mathcal{G}) \right] \right] + \lambda \cdot \|q_\phi(\mathcal{G})\|_1,$$

where $\phi$ and $\theta$ are the parameters of the causal mask predictor and the dynamics model, respectively, $\lambda$ is a regularization coefficient, $\tau$ represents a trajectory sampled from dataset $\mathcal{D}$, and $\mathcal{G}$ denotes the causal graph. Different from Varambally et al. (2024), which can infer a causal graph from a single sample (i.e., $q_\phi(\mathcal{G}) := q_\phi(\mathcal{G} \mid X)$), our focus is on the sequential decision problem where the causal model is used to generate trajectories. To decompose superposed causal relationships in offline trajectories, we use a trajectory-wise inference model for causal graph prediction $q_\phi(\mathcal{G}) := q_\phi(\mathcal{G} \mid \tau)$ and a transition model for trajectory reconstruction $p_\theta(\tau \mid \mathcal{G}) = \prod_{t=0}^T p_\theta(s_{t+1} \mid s_t, a_t, \mathcal{G})$. Then we have the following objective:

$$\min_{\phi,\theta} -\mathbb{E}_{\tau \sim \mathcal{D}} \left[ \mathbb{E}_{q_\phi(\mathcal{G}|\tau)} \left[ \sum_{t=0}^T \log p_\theta(s_{t+1} \mid s_t, a_t, \mathcal{G}) \right] + \lambda \cdot \|\mathcal{G}\|_1 \right]. \tag{2}$$

To optimize Eq. 2, we design an simple yet efficient end-to-end architecture based on Transformers. The network includes the following modules: (1) a causal mask predictor $q_\phi$ that utilizes a Transformer to take trajectories as input and outputs the distribution $q_\phi(\mathcal{G} \mid \tau)$; (2) a fully connected

feature encoder $f_{\theta_1}$ that encodes single time-step states and actions; and (3) a dynamics model $f_{\theta_2}$ that predicts the next state.

To predict $s_{t+1}$ given the previous trajectory $\tau$, we first use the Transformer-based causal mask predictor $q_\phi(\mathcal{G} \mid \tau)$ to obtain the distribution of the causal mask $\mathcal{G}$. We then sample $\mathcal{G} \sim q_\phi(\mathcal{G} \mid \tau)$. After obtaining the causal mask, we predict $s_{t+1}$ based on $s_t$ and $a_t$ with the causal mask by computing $f_{\theta_2}(s_{t+1}|f_{\theta_1}(s_t, a_t) \circ \mathcal{G}, \mathrm{sg}(\mathcal{G}))$, where $f_{\theta_1}(s_t, a_t) \circ \mathcal{G}$ represents masking unrelated features using $\mathcal{G}$ and $\mathrm{sg}(\mathcal{G})$ is an appended input of causal mask which the gradient will be stopped when optimized. In summary, we have $q_\phi(\mathcal{G} \mid \tau) := f_{\theta_2}(s_{t+1}|f_{\theta_1}(s_t, a_t) \circ \mathcal{G}, \mathrm{sg}(\mathcal{G}))$. To enable sampling of discrete values $\{0, 1\}$ through $q_\phi(\mathcal{G} \mid \tau)$ while keeping the entire process differentiable, we utilize the Gumbel-Softmax trick (Maddison et al., 2017) to implement the distribution sampling process $\mathcal{G} \sim q_\phi(\mathcal{G} \mid \tau)$. The network is trained end-to-end by Eq. 2 and we called the whole training method **S**uperposed c**A**usal **M**odel (SAM) learning.

## 5 EXPERIMENTS

In our experiments, we build two new benchmarks for these settings, namely Mixed-Chemical and Confusing-Minigrid, which are introduced in Section 5.1. We primarily investigate the following research questions:

1. **RQ1**: Is SAM capable of inferring superposed causal relationships from offline sequential decision-making data with multiple causal relationships (see Section 5.2)?

2. **RQ2**: Does superposed causal graph learning simplify the learning transition and enhance generalizability compared to single-graph models or dense models (see Sections 5.3)?

3. **RQ3**: Can SAM identify causal relationships encountered by the agent in real time to facilitate better decision-making (see Section 5.4)?

### 5.1 EXPERIMENT SETTINGS

Most reinforcement learning environments lack explicit causal graphs for generalization purposes. To evaluate our methods, we constructed benchmarks based on the widely recognized Chemical environment (Ke et al., 2021) and a custom-designed Minigrid environment Chevalier-Boisvert et al. (2023) as shown in Figure 2. For each environment, we evaluate the model under two settings: the target setting and the spurious setting. In the target setting, the policy aims to perform on one of the latent causal tasks with the model trained on the superposed causal dataset. In the spurious setting, the policy is tested in an environment with noise on spurious correlation, as shown in the following, which is the key setting to demonstrate the generalization ability. We conducted experiments in the offline setting, where each environment encompasses multiple settings of latent causal relationships. Our dataset comprises 200,000 transition steps for each environment.

1. **Mixed-Chemical**: The Chemical environment (Ke et al., 2021) consists of 10 colored nodes, where color changes are governed by an underlying causal graph. At each step, the agent changes the color of one node, causing its descendant nodes in the causal graph to change color sequentially. The goal is to change all node colors to a desired target color. We modified the environment to include nine randomly generated, unknown causal graphs, transforming it into a multi-causal graph reinforcement learning environment. Data collection was performed using a random policy, generating datasets in the offline setting. In the spurious setting, the agent receives noisy observations for certain nodes. By disentangling the superposed causal graphs, the agent can ignore spurious correlations and improve generalization.

2. **Confusing-Minigrid**: Minigrid Chevalier-Boisvert et al. (2023) is a grid-based environment where the agent aims to navigate around obstacles to reach a target grid in as few steps as possible. We introduced multiple latent causal graphs that determine whether obstacles can be moved or covered with the agent and other obstacles, resulting in diverse environmental transitions. We name the tasks with the format of $M\{?\} - C\{?\}$ to denote the properties of the obstacles. This transforms the environment into a multi-causal graph reinforcement learning setting. Given the complexity of the state space, we utilized an early-stopping Proximal Policy Optimization (PPO) algorithm for data collection. In

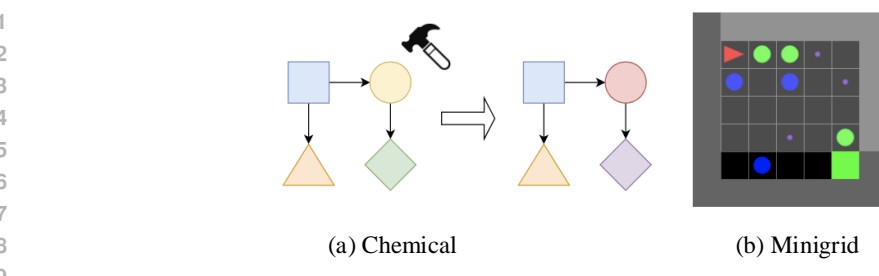

(a) Chemical                    (b) Minigrid

Figure 2: Illustration of the Chemical and Minigrid Environments.

the spurious setting, obstacles that do not covered others are initialized in wider positions compared to those in the training dataset. Accurately identifying the environmental causal relationships is crucial for moving obstacles that block the path and helping the agent reach the target via the shortest route.

**Baselines** To fairly compare our methods, We implement the following baselines: **Modular** (Ke et al. (2021)): predicts each state variable with a separate network. **GNN** (Kipf et al. (2020)): implements graph neural network that learns relational information. **kmeans + CDL**: CDL infers a single causal structure for the entire dataset to learn dynamics. To learn superposed causal structure, we cluster trajectories with time series cluster Tavenard et al. (2020) and train each group with CDL (Wang et al., 2022). **FCDL** (Hwang et al. (2024)): quantizes the state-action spaces into subgroups and learns fine-grained causal structure. However, we identify causal graphs through transitions between states rather than single current state observations, making it more efficient for unobserved variables. **Oracle**: Utilizes the same neural network architecture as ours but with the ground truth causal graph, serving as an upper bound for SAM.

5.2    ACCURACY OF CAUSAL RELATIONSHIP DISCOVERY

Tables 1 and 2 present the Structural Hamming Distance (SHD) used to evaluate our causal discovery methods. We compare our proposed method, SAM, specifically with KMeans+CDL and FCDL, as other methods do not explicitly focus on causal graph discovery.

Table 1: Causal-relationship accuracy on Mixed-Chemical environment, where we **bold** the best result performance for each task.

| Method | random0 | random1 | random2 | random3 | random4 | random5 | random6 | random7 | random8 | Average |
|---|---|---|---|---|---|---|---|---|---|---|
| kmeans+CDL | 27.35 | 29.35 | 39.87 | 40.73 | 28.52 | 35.88 | 35.57 | 32.42 | 30.30 | 33.33 |
| FCDL | 22.93 | 21.30 | 35.30 | 32.19 | 25.14 | 30.49 | 26.70 | 23.07 | 26.38 | 27.06 |
| SAM | **8.16** | **3.91** | **0.52** | **2.05** | **4.45** | **3.66** | **5.31** | **5.10** | **3.28** | **4.05** |

Table 2: Causal-relationship accuracy on Confusing-Maze environment, where we **bold** the best result performance for each task.

| Method | M-C012 | M01-C0 | M01-C2 | M012-C012 | M012-C | M02-C1 | Average |
|---|---|---|---|---|---|---|---|
| kmeans+CDL | 3.54 | 9.88 | 9.83 | **0.53** | 15.95 | 9.90 | 8.27 |
| FCDL | 25.00 | 15.00 | 15.00 | 22.00 | 7.00 | 15.00 | 16.50 |
| SAM | **0.00** | **7.11** | **5.24** | 3.28 | **9.74** | **8.28** | **5.61** |

Our method, SAM, demonstrates superior performance across all datasets. In the Mixed-Chemical environment (Table 1), SAM achieves an average SHD of **4.05**, significantly lower than KMeans+CDL and FCDL, which have average SHDs of **33.33** and **27.06**, respectively. Notably, SAM consistently attains the lowest SHD in all nine random graphs, with SHD values ranging from **0.52** to **8.16**, indicating its high accuracy in reconstructing the underlying causal graphs. In the Confusing-Minigrid environment (Table 2), SAM also outperforms the other methods, achieving an

average SHD of **5.61** compared to **8.27** for KMeans+CDL and **16.50** for FCDL. SAM attains the lowest SHD in five out of six settings, demonstrating its robustness and effectiveness in more complex environments. For instance, in the M-C012 setting, SAM achieves an SHD of **0.00**, perfectly recovering the causal graph, whereas KMeans+CDL and FCDL obtain SHDs of **3.54** and **25.00**, respectively.

The poor performance of KMeans+CDL can be attributed to its inability to effectively cluster data points that share the same causal relationships in high-dimensional spaces, leading to inaccurate causal graph estimations. Although FCDL yields better results than KMeans+CDL, it struggles to distinguish superposed causal graphs from time-series data, limiting its ability to accurately infer the underlying causal structures.

In contrast, SAM excels at disentangling superposed causal graphs and accurately inferring causal relationships from sequential decision-making data. Its superior performance is evident across different environments and settings, highlighting its advantage in handling multiple latent causal relationships. This leads to more precise causal discovery and enhances the generalization capabilities of the agent in diverse and complex environments.

### 5.3 Prediction Error of the Learned World Models

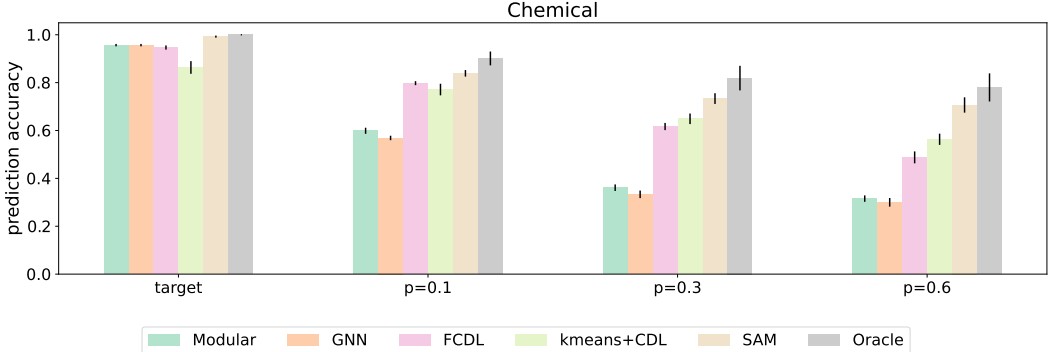

Figure 3: Chemical prediction accuracy with different noise under target setting and spurious setting. The variable p on the axis represents the probability of adding noise to a node in spurious setting. Accuracy represents the correctness of color prediction on clean nodes.

To demonstrate that learning multiple causal graphs improves generalization, we evaluate the prediction accuracy on the Mixed-Chemical environment under both target and spurious settings. In the spurious setting, noise is introduced to the nodes with probabilities of *0.1, 0.3, and 0.6*. The results are shown in Figure 3. We do not include the results of Confusing-Minigrid environment to this section of analyzing because of the scale of each dimension in state space are different, which is meaningless by demonstrating their average performance.

In the target setting, all methods perform well, as expected given the simplicity of the environment. However, in the spurious settings, where noise is added, we observe substantial performance degradation, particularly for models without explicit causal mechanisms. The single causal-relationship world methods (such as Modular and GNN) shows the most significant drop, confirming its limited ability to handle noisy and complex causal structures.

Among the methods incorporating causal learning, SAM clearly outperforms the baselines across all spurious settings. For instance, at noise probability $p = 0.3$, SAM achieves substantially higher prediction accuracy than KMeans+CDL and FCDL. While the Modular and GNN approaches capture some relational information, their generalization capabilities diminish as noise increases, highlighting their limitations in disentangling superposed causal structures. At $p = 0.6$, SAM maintains robust performance, significantly narrowing the gap with the Oracle, which represents the upper bound using the true causal graph. This result emphasizes the strength of SAM in leveraging latent causal graphs to generalize effectively, even under high levels of noise. SAM's close performance to the Oracle, particularly at higher noise levels, demonstrates its efficacy in handling both observed and unobserved variables within complex, noisy environments.

Overall, the results in Figure 3 strongly support the effectiveness of SAM. By accurately identifying and disentangling superposed causal graphs, SAM achieves superior generalization performance compared to all other methods, including state-of-the-art baselines such as GNN and FCDL. This underscores SAM's significant advantage in environments with complex, overlapping causal structures and noisy observations. The detailed results of each task are shown in Appendix B.

## 5.4 Performance of Model Predictive Controls

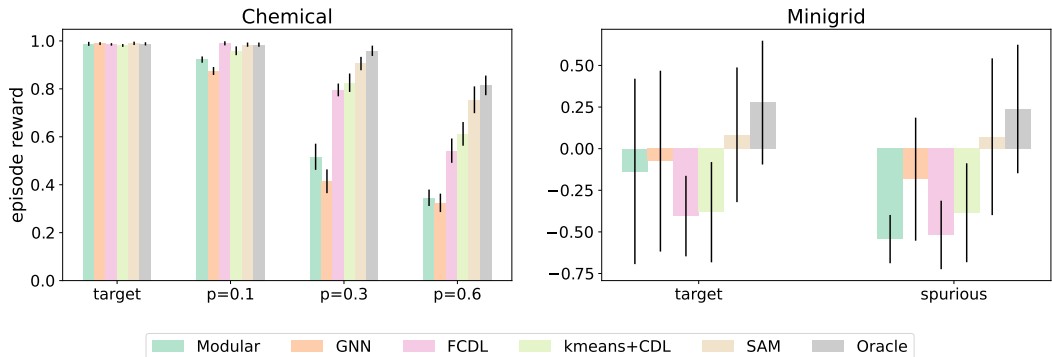

Figure 4: **Left:** Performance of model predictive controls in the chemical environment under target and spurious settings (probability of noise: 0.1, 0.3, 0.6). **Right:** Performance of model predictive controls in the Minigrid environment under target and spurious settings.

We evaluate the testing rewards of all methods across various tasks, with the summarized results presented in Figure 4. Detailed results for each individual task can be found in Appendix B.

In the target setting of the Mixed-Chemical environment (left plot of Figure 4), the task's simplicity allows all methods to achieve high performance, as expected. However, the true challenge arises in the spurious settings, where noise is introduced. At noise levels of $p = 0.1$, $p = 0.3$, and $p = 0.6$, our method, SAM, consistently outperforms the baseline methods, demonstrating significantly higher episode rewards. Notably, at the highest noise level $p = 0.6$, SAM maintains robust performance, whereas other methods, particularly the single causal-relationship world models like Modular and GNN, experience substantial performance degradation.

In the Confusing-Minigrid environment (right plot of Figure 4), the increased complexity, caused by compounded causal effects, makes learning more challenging for all models. Despite this, SAM again outperforms all other methods in both target and spurious settings. The performance drop of the baselines in the spurious setting is evident, as they fail to effectively discard spurious correlations introduced by noise. SAM, however, mitigates these challenges, leveraging its causal graph learning capabilities to better generalize across tasks. In the target setting, SAM performs comparably to the Oracle, indicating its ability to approximate the true causal structure even in complex environments.

In the spurious setting, baseline methods show significant performance degradation, particularly those relying on single causal-relationship world relational learning (e.g., GNN and Modular), as they are more susceptible to overfitting to spurious correlations. Causal models such as FCDL perform better than single-causal-relationship discovery methods, but still suffer from reduced generalization due to their inability to effectively handle overlapping causal structures. In contrast, SAM excels in this environment, as it successfully disentangles superposed causal graphs, allowing it to ignore irrelevant correlations and maintain stable performance across tasks.

Overall, the results in Figure 4 highlight the advantages of SAM, particularly its ability to generalize in noisy and complex environments. By leveraging causal discovery and focusing on relevant causal structures, SAM consistently outperforms baseline methods and approaches the performance of the Oracle, even in the most challenging settings.

## 5.5 ABLATION STUDIES

To demonstrate that the superior generalization of our method is not solely due to temporal properties utilized by the network structure but rather due to the accurate identification of causal graphs, we conduct an ablation study comparing our method (SAM) with the RNN baseline (Figure 5).

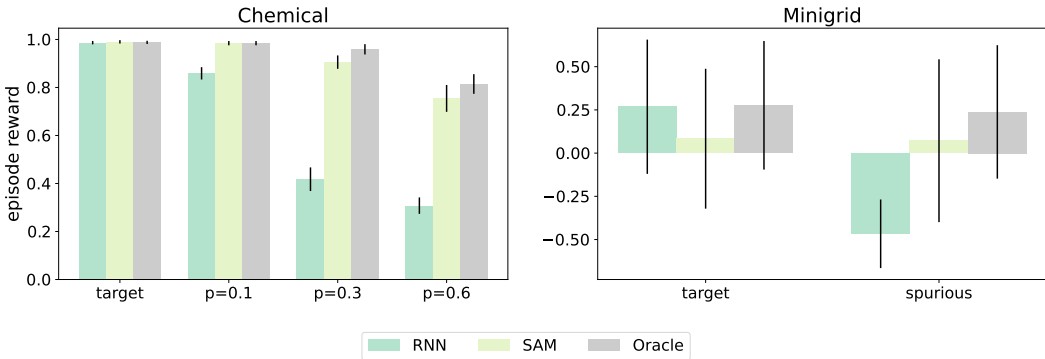

Figure 5: **Left:** Performance of model predictive controls in the chemical environment under target and spurious settings (probability of noise: 0.1, 0.3, 0.6). **Right:** Performance of model predictive controls in the Minigrid environment under target and spurious settings.

In the **target setting**, RNN performs similarly to or even slightly better than SAM due to the simplicity of the task. However, in the **spurious setting**, where noise is introduced, SAM significantly outperforms RNN. As noise increases ($p = 0.1$, $p = 0.3$, and $p = 0.6$), RNN's performance drops sharply, while SAM maintains high rewards. This demonstrates that RNN struggles to handle noise due to its reliance on temporal patterns, which are disrupted by spurious correlations.

In contrast, SAM's robustness in noisy environments shows that its generalization ability is due to accurate causal graph identification, not temporal dependencies. In the Confusing-Minigrid environment, we see a similar trend, with SAM consistently outperforming RNN, especially in the spurious setting.

Overall, the results clearly show that SAM's strength lies in its ability to disentangle causal structures, leading to superior performance in complex and noisy environments.

## 5.6 VISUALIZATION OF ON-THE-FLY CAUSAL-RELATIONSHIP DISCOVERY IN SAM

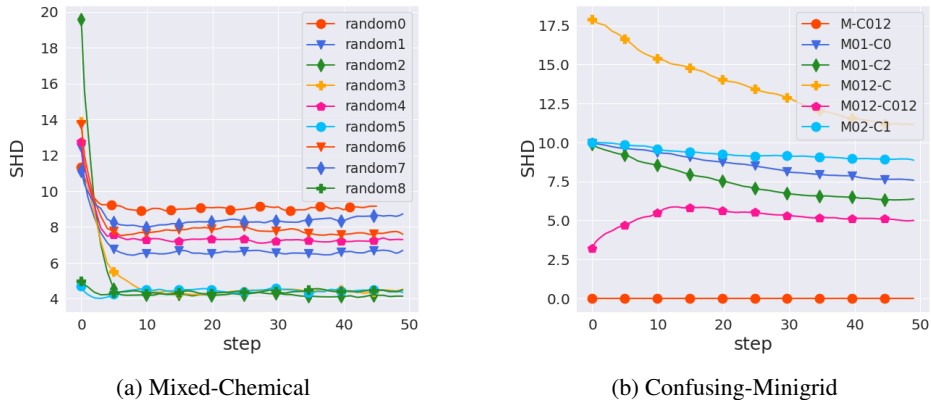

(a) Mixed-Chemical

(b) Confusing-Minigrid

Figure 6: Structural Hamming Distance (SHD) over time steps in (a) the Mixed-Chemical environment and (b) the Confusing-Minigrid environment, demonstrating SAM's adaptation to the true causal graph.

We demonstrate that SAM can dynamically adapt to the true causal relationships over time steps. To illustrate this capability, we conducted experiments where each target task runs for 50 episodes, and

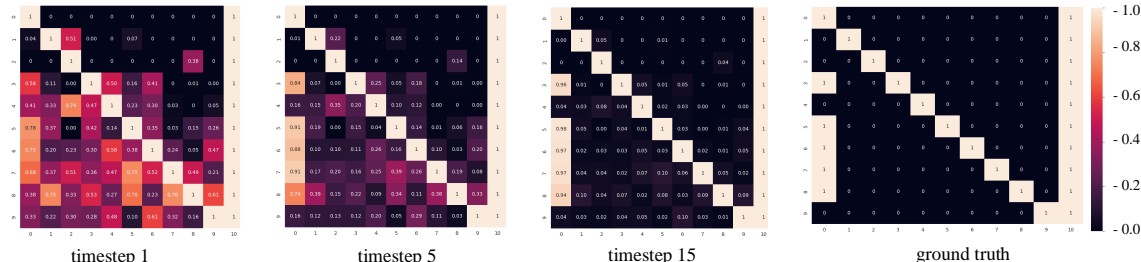

Figure 7: Illustration of Causal Graph Adaptation Over Time. The inferred causal graph gradually aligns with the ground truth as time progresses.

we report the average Structural Hamming Distance (SHD) at each time step (Figure 6). As shown in Figure 6, the SHD decreases over time in both the Mixed-Chemical and Confusing-Mi'ni'gri'd environments, indicating that the inferred causal graph progressively aligns with the ground truth. In the **Mixed-Chemical** environment (Figure 6a), SAM quickly adapts, with SHD values stabilizing after approximately 10 steps. This rapid convergence demonstrates the method's efficiency in learning the underlying causal structure even when faced with complex, superposed causal graphs. In the **Confusing-Minigrid** environment (Figure 6b), SAM continues to reduce SHD over time, though at a slightly slower rate due to the higher complexity of the causal relationships in this environment. Despite this complexity, SAM steadily improves, further proving its robustness in diverse settings. Importantly, in both environments, SAM consistently outperforms competing methods in terms of how quickly and accurately it can discover the true causal relationships.

Additionally, Figure 7 provides a visual example of how SAM's inferred causal graph evolves over time. At earlier timesteps (e.g., timestep 1), the inferred graph is far from the ground truth, but by timestep 15, it closely aligns with the true causal structure, demonstrating SAM's ability to refine its understanding of the causal relationships through sequential data. By timestep 50, SAM's inferred graph is nearly identical to the ground truth, showing its capability for accurate and adaptive causal discovery.

These results confirm that SAM can efficiently and accurately discover and adapt causal relationships in real-time, making it a highly effective approach in complex, dynamic environments where causal structures evolve over time.

## 6 CONCLUSION AND DISCUSSION

In this paper, we introduced **S**uperposed c**A**usal **M**odel (SAM), a novel approach for learning superposed causal world models that can handle multiple, dynamic causal relationships in sequential decision-making environments. SAM addresses the limitations of existing causal world models, which assume a single causal structure governs the entire dataset. By allowing for the identification of distinct causal relationships across different episodes, SAM significantly improves generalization, especially in heterogeneous environments where causal dynamics vary across contexts. Through extensive experiments in environments like Mixed-Chemical and Confusing-Maze, we demonstrated that SAM effectively learns and adapts causal relationships in real-time. Our results show that SAM outperforms existing baselines, particularly in challenging spurious settings where noise and multi-modal causal structures obscure traditional models' performance. Unlike dense models that struggle with redundant relationships and causal models that cannot handle mixtures of causal graphs, SAM robustly disentangles overlapping causal structures, leading to superior prediction accuracy and decision-making performance across various tasks.

SAM's ability to generalize across complex and evolving environments suggests its potential for real-world applications in areas like dynamic resource allocation and autonomous systems. Despite its promising results, SAM has some limitations, especially that SAM's computational cost increases with the complexity of the environment due to the use of a Transformer-based architecture. Future work could explore scaling SAM to larger datasets, further improving its efficiency, and investigating its applicability in broader decision-making contexts where causal structures are even more diverse and challenging.

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

## A DERIVATION OF ELBO

For offline dataset $\mathcal{D} = \{\tau_i\}_{i=1}^N$ and causal graphs $\mathcal{G}_{1:C} = (\mathcal{G}_1, ..., \mathcal{G}_C)$, following Varambally et al. (2024), log-likelihood can be written as follow:

$$\log p_\theta(\tau) = \log[\sum_{\mathcal{G}_{1:C}} p_\theta(\tau|\mathcal{G}_{1:C})p(\mathcal{G}_{1:C}) \times \frac{q_\phi(\mathcal{G}_{1:C})}{q_\phi(\mathcal{G}_{1:C})}]$$

$$= \log \mathbb{E}_{q_\phi(\mathcal{G}_{1:C})} \left[ \frac{p_\theta(\tau \mid \mathcal{G}_{1:C}) p(\mathcal{G}_{1:C})}{q_\phi(\mathcal{G}_{1:C})} \right]$$

$$\geq \mathbb{E}_{q_\phi(\mathcal{G}_{1:C})} \left[ \log \frac{p_\theta(\tau \mid \mathcal{G}_{1:C}) p(\mathcal{G}_{1:C})}{q_\phi(\mathcal{G}_{1:C})} \right] \quad \text{(using Jensen's inequality)}$$

$$= \mathbb{E}_{q_\phi(\mathcal{G}_{1:C})} [\log p_\theta(\tau \mid \mathcal{G}_{1:C})] - D_{\mathrm{KL}}(q_\phi(\mathcal{G}_{1:C}) \parallel p(\mathcal{G}_{1:C}))$$

Given that each trajectory is conditionally independent under the causal models, we can express this as:

$$\mathbb{E}_{\tau \sim \mathcal{D}}[\log p_\theta(\tau)] \geq \mathbb{E}_{\tau \sim \mathcal{D}}[\mathbb{E}_{q_\phi(\mathcal{G}_{1:C})} [\log p_\theta(\tau \mid \mathcal{G}_{1:C})] - D_{\mathrm{KL}}(q_\phi(\mathcal{G}_{1:C}) \parallel p(\mathcal{G}_{1:C}))]$$

KL divergence between $\log p(\mathcal{G}_i)$, $\log q_\phi(\mathcal{G}_i)$ can formulate as sparsity regularization based on the following proposition.

**Proposition 1** (KL Divergence as Sparsity Regularization). *With entry-wise independent Bernoulli prior $p(\mathcal{G})$ and point mass variational distribution $q(\mathcal{G} \mid \tau)$ of $DAGs$, $\mathbb{D}_{KL}[q_\phi\|p]$ is equivalent to an $\ell_1$ sparsity regularization for the discovered causal graph. Ding et al. (2022)*

Applying sparsity regularization, we have the loss function

$$\min_{\phi,\theta} -\mathbb{E}_{\tau \sim \mathcal{D}}[\mathbb{E}_{q_\phi(\mathcal{G}_{1:C})} [\log p_\theta(\tau \mid \mathcal{G}_{1:C})] + \lambda \cdot \|q_\phi(\mathcal{G}_{1:C})\|_1 \tag{3}$$

## B DETAIL RESULT OF ENVIRONMENTS

Table 3: Detailed results of "target" setting reward for each task in Mixed-Chemical environment, where "()" denote the standard deviation among three seeds.

| Environment | Modular | GNN | RNN | FCDL | kmeans+CDL | Oracle | SAM |
|---|---|---|---|---|---|---|---|
| Chemical-random0 | 0.99(0.03) | 0.99(0.04) | 0.99(0.05) | 0.99(0.04) | 0.98(0.05) | 0.99(0.03) | 0.99(0.03) |
| Chemical-random1 | 0.99(0.03) | 0.99(0.02) | 0.99(0.04) | 0.99(0.03) | 0.97(0.07) | 0.99(0.03) | 1.00(0.02) |
| Chemical-random2 | 0.98(0.06) | 0.99(0.02) | 0.99(0.03) | 0.98(0.07) | 0.98(0.07) | 0.98(0.04) | 0.99(0.03) |
| Chemical-random3 | 0.99(0.03) | 0.99(0.03) | 0.99(0.03) | 0.99(0.04) | 0.98(0.05) | 0.99(0.03) | 1.00(0.01) |
| Chemical-random4 | 0.99(0.04) | 0.98(0.04) | 0.97(0.08) | 0.99(0.04) | 0.98(0.06) | 0.99(0.05) | 0.99(0.04) |
| Chemical-random5 | 0.99(0.03) | 0.99(0.04) | 0.99(0.04) | 0.99(0.03) | 0.99(0.06) | 1.00(0.02) | 0.99(0.04) |
| Chemical-random6 | 0.99(0.03) | 0.99(0.03) | 0.99(0.04) | 0.98(0.05) | 0.99(0.03) | 0.98(0.06) | 0.98(0.05) |
| Chemical-random7 | 0.97(0.09) | 0.98(0.03) | 0.98(0.06) | 0.98(0.06) | 0.98(0.07) | 0.98(0.05) | 0.98(0.06) |
| Chemical-random8 | 1.00(0.02) | 1.00(0.02) | 0.99(0.04) | 0.98(0.05) | 0.98(0.06) | 0.99(0.03) | 0.99(0.03) |

Table 4: Detailed results of "target" setting reward for each task in Confusing-Minigrid environment, where "()" denote the standard deviation among three seeds.

| Environment | Modular | GNN | RNN | FCDL | kmeans+CDL | Oracle | SAM |
|---|---|---|---|---|---|---|---|
| Minigrid-M-C012 | 0.46(0.44) | 0.52(0.27) | 0.76(0.09) | -0.14(0.26) | 0.00(0.18) | 0.75(0.11) | 0.57(0.28) |
| Minigrid-M01-C0 | -0.48(0.47) | -0.35(0.55) | -0.01(0.55) | -0.57(0.25) | -0.55(0.31) | 0.09(0.55) | -0.23(0.44) |
| Minigrid-M01-C2 | -0.44(0.48) | -0.44(0.50) | 0.12(0.49) | -0.49(0.29) | -0.55(0.29) | 0.21(0.54) | -0.14(0.45) |
| Minigrid-M012-C012 | 0.68(0.24) | 0.69(0.12) | 0.76(0.05) | -0.06(0.26) | 0.00(0.14) | 0.70(0.23) | 0.60(0.24) |
| Minigrid-M012-C | -0.63(0.37) | -0.62(0.37) | -0.10(0.45) | -0.63(0.25) | -0.68(0.25) | -0.19(0.44) | -0.32(0.37) |
| Minigrid-M02-C1 | -0.41(0.55) | -0.25(0.60) | 0.08(0.51) | -0.54(0.31) | -0.51(0.34) | 0.10(0.53) | 0.02(0.51) |

Table 5: Detailed results of "prob-0.1" setting reward for each task in Mixed-Chemical environment, where "()" denote the standard deviation among three seeds.

| Environment | Modular | GNN | RNN | FCDL | kmeans+CDL | Oracle | SAM |
|---|---|---|---|---|---|---|---|
| Chemical-random0 | 0.94(0.14) | 0.90(0.18) | 0.89(0.20) | 0.99(0.06) | 0.94(0.13) | 0.99(0.04) | 0.99(0.06) |
| Chemical-random1 | 0.93(0.15) | 0.87(0.21) | 0.88(0.21) | 0.99(0.06) | 0.95(0.11) | 1.00(0.01) | 1.00(0.04) |
| Chemical-random2 | 0.91(0.18) | 0.88(0.21) | 0.83(0.26) | 0.99(0.05) | 0.96(0.11) | 0.98(0.06) | 0.98(0.08) |
| Chemical-random3 | 0.91(0.17) | 0.86(0.21) | 0.83(0.25) | 1.00(0.00) | 0.94(0.11) | 0.99(0.03) | 0.98(0.07) |
| Chemical-random4 | 0.91(0.18) | 0.86(0.23) | 0.82(0.25) | 1.00(0.02) | 0.96(0.11) | 0.97(0.09) | 0.99(0.04) |
| Chemical-random5 | 0.94(0.15) | 0.85(0.22) | 0.86(0.22) | 0.99(0.04) | 0.94(0.13) | 0.98(0.06) | 0.98(0.08) |
| Chemical-random6 | 0.93(0.16) | 0.87(0.21) | 0.88(0.21) | 0.99(0.04) | 0.98(0.06) | 0.98(0.08) | 0.97(0.10) |
| Chemical-random7 | 0.91(0.18) | 0.89(0.22) | 0.87(0.22) | 0.97(0.12) | 0.99(0.04) | 0.98(0.05) | 0.98(0.08) |
| Chemical-random8 | 0.92(0.17) | 0.89(0.21) | 0.87(0.21) | 0.99(0.05) | 0.97(0.09) | 0.99(0.04) | 0.99(0.07) |

Table 6: Detailed results of "prob-0.3" setting reward for each task in Mixed-Chemical environment, where "()" denote the standard deviation among three seeds.

| Environment | Modular | GNN | RNN | FCDL | kmeans+CDL | Oracle | SAM |
|---|---|---|---|---|---|---|---|
| Chemical-random0 | 0.44(0.23) | 0.40(0.21) | 0.39(0.20) | 0.80(0.23) | 0.78(0.25) | 0.94(0.12) | 0.95(0.14) |
| Chemical-random1 | 0.45(0.25) | 0.37(0.25) | 0.39(0.19) | 0.84(0.20) | 0.80(0.26) | 0.97(0.10) | 0.91(0.18) |
| Chemical-random2 | 0.56(0.31) | 0.47(0.28) | 0.47(0.29) | 0.76(0.26) | 0.83(0.23) | 0.98(0.08) | 0.88(0.20) |
| Chemical-random3 | 0.54(0.24) | 0.40(0.22) | 0.39(0.21) | 0.80(0.23) | 0.81(0.23) | 0.99(0.05) | 0.91(0.17) |
| Chemical-random4 | 0.50(0.26) | 0.35(0.23) | 0.38(0.25) | 0.77(0.25) | 0.81(0.25) | 0.94(0.14) | 0.94(0.13) |
| Chemical-random5 | 0.51(0.23) | 0.39(0.22) | 0.37(0.24) | 0.80(0.22) | 0.81(0.21) | 0.98(0.06) | 0.90(0.16) |
| Chemical-random6 | 0.59(0.27) | 0.47(0.26) | 0.46(0.26) | 0.83(0.21) | 0.86(0.21) | 0.95(0.14) | 0.89(0.19) |
| Chemical-random7 | 0.58(0.31) | 0.49(0.28) | 0.51(0.32) | 0.78(0.26) | 0.91(0.18) | 0.93(0.14) | 0.86(0.24) |
| Chemical-random8 | 0.48(0.24) | 0.39(0.25) | 0.40(0.25) | 0.78(0.25) | 0.82(0.24) | 0.95(0.12) | 0.91(0.17) |

Table 7: Detailed results of "prob-0.6" setting reward for each task in Mixed-Chemical environment, where "()" denote the standard deviation among three seeds.

| Environment | Modular | GNN | RNN | FCDL | kmeans+CDL | Oracle | SAM |
|---|---|---|---|---|---|---|---|
| Chemical-random0 | 0.32(0.20) | 0.28(0.19) | 0.28(0.18) | 0.51(0.26) | 0.62(0.24) | 0.85(0.20) | 0.82(0.23) |
| Chemical-random1 | 0.31(0.19) | 0.29(0.17) | 0.27(0.19) | 0.58(0.25) | 0.59(0.25) | 0.77(0.23) | 0.77(0.23) |
| Chemical-random2 | 0.38(0.22) | 0.38(0.29) | 0.33(0.24) | 0.59(0.29) | 0.64(0.30) | 0.83(0.22) | 0.69(0.28) |
| Chemical-random3 | 0.36(0.19) | 0.33(0.20) | 0.29(0.17) | 0.48(0.21) | 0.57(0.24) | 0.87(0.19) | 0.74(0.24) |
| Chemical-random4 | 0.31(0.20) | 0.32(0.23) | 0.31(0.23) | 0.51(0.29) | 0.59(0.27) | 0.80(0.24) | 0.80(0.24) |
| Chemical-random5 | 0.35(0.21) | 0.29(0.17) | 0.29(0.16) | 0.49(0.23) | 0.55(0.24) | 0.86(0.20) | 0.83(0.22) |
| Chemical-random6 | 0.37(0.24) | 0.35(0.21) | 0.33(0.19) | 0.56(0.26) | 0.67(0.26) | 0.80(0.21) | 0.69(0.25) |
| Chemical-random7 | 0.40(0.29) | 0.38(0.25) | 0.38(0.25) | 0.63(0.27) | 0.70(0.29) | 0.75(0.27) | 0.69(0.29) |
| Chemical-random8 | 0.31(0.17) | 0.30(0.20) | 0.29(0.17) | 0.53(0.27) | 0.58(0.28) | 0.80(0.24) | 0.76(0.23) |

Table 8: Detailed results of "spurious" setting reward for each task in Confusing-Minigrid environment, where "()" denote the standard deviation among three seeds.

| Environment | Modular | GNN | RNN | FCDL | kmeans+CDL | Oracle | SAM |
|---|---|---|---|---|---|---|---|
| Minigrid-M-C012 | -0.42(0.18) | 0.30(0.47) | -0.61(0.27) | -0.51(0.24) | -0.03(0.01) | 0.75(0.11) | 0.70(0.14) |
| Minigrid-M01-C0 | -0.45(0.46) | -0.39(0.54) | -0.46(0.45) | -0.54(0.32) | -0.58(0.31) | -0.03(0.51) | -0.38(0.33) |
| Minigrid-M01-C2 | -0.79(0.19) | -0.39(0.49) | -0.49(0.41) | -0.62(0.18) | -0.53(0.29) | 0.11(0.52) | -0.08(0.49) |
| Minigrid-M012-C012 | -0.43(0.24) | 0.23(0.38) | -0.67(0.21) | -0.12(0.39) | 0.01(0.10) | 0.68(0.23) | 0.62(0.28) |
| Minigrid-M012-C | -0.63(0.37) | -0.62(0.37) | -0.10(0.45) | -0.63(0.25) | -0.68(0.25) | -0.19(0.44) | -0.32(0.37) |
| Minigrid-M02-C1 | -0.54(0.34) | -0.23(0.53) | -0.47(0.40) | -0.69(0.20) | -0.50(0.32) | 0.11(0.51) | -0.11(0.47) |

## C   EXPERIMENTAL DETAILS

### C.1   ENVIRONMENT DETAILS

In a two-dimensional grid world environment, there is an agent, a target, and multiple colored obstacles. Obstacles of the same color possess identical mobility and obstructive properties, which reflect the causality of the environment. For instance, if an obstacle does not impede other obstacles from the perspective of the agent, then the position of that obstacle is causally unrelated to the positions of other obstacles. Conversely, if an obstacle can be moved, then there is a causal relationship between

Table 9: ood policy mpe for each env

| Environment | Modular | GNN | RNN | FCDL | kmeans+CDL | Oracle | SAM |
|---|---|---|---|---|---|---|---|
| Chemical-random0 | 0.95(0.22) | 0.95(0.21) | 0.88(0.32) | 0.93(0.25) | 0.84(0.37) | 1.00(0.05) | 0.99(0.11) |
| Chemical-random1 | 0.95(0.22) | 0.95(0.21) | 0.89(0.31) | 0.95(0.23) | 0.84(0.37) | 1.00(0.05) | 0.99(0.09) |
| Chemical-random2 | 0.96(0.20) | 0.96(0.20) | 0.92(0.28) | 0.95(0.21) | 0.91(0.28) | 1.00(0.00) | 0.99(0.09) |
| Chemical-random3 | 0.96(0.20) | 0.96(0.20) | 0.90(0.30) | 0.95(0.22) | 0.87(0.34) | 1.00(0.01) | 0.99(0.09) |
| Chemical-random4 | 0.96(0.21) | 0.95(0.21) | 0.89(0.31) | 0.94(0.23) | 0.85(0.36) | 1.00(0.03) | 0.99(0.08) |
| Chemical-random5 | 0.95(0.21) | 0.96(0.21) | 0.91(0.29) | 0.94(0.23) | 0.85(0.35) | 1.00(0.04) | 0.99(0.08) |
| Chemical-random6 | 0.96(0.19) | 0.96(0.19) | 0.92(0.27) | 0.96(0.20) | 0.87(0.34) | 1.00(0.02) | 1.00(0.07) |
| Chemical-random7 | 0.96(0.20) | 0.96(0.20) | 0.91(0.29) | 0.95(0.22) | 0.90(0.30) | 1.00(0.03) | 0.99(0.08) |
| Chemical-random8 | 0.96(0.20) | 0.96(0.20) | 0.90(0.30) | 0.95(0.22) | 0.84(0.37) | 1.00(0.02) | 1.00(0.06) |

Table 10: ood noise prob0.1 mpe for each env

| Environment | Modular | GNN | RNN | FCDL | kmeans+CDL | Oracle | SAM |
|---|---|---|---|---|---|---|---|
| Chemical-random0 | 0.58(0.49) | 0.56(0.50) | 0.55(0.50) | 0.79(0.41) | 0.75(0.44) | 0.87(0.34) | 0.84(0.37) |
| Chemical-random1 | 0.58(0.49) | 0.58(0.49) | 0.56(0.50) | 0.81(0.40) | 0.75(0.43) | 0.88(0.33) | 0.86(0.35) |
| Chemical-random2 | 0.60(0.49) | 0.57(0.50) | 0.55(0.50) | 0.79(0.41) | 0.81(0.39) | 0.94(0.24) | 0.82(0.38) |
| Chemical-random3 | 0.61(0.49) | 0.57(0.50) | 0.54(0.50) | 0.79(0.41) | 0.77(0.42) | 0.95(0.22) | 0.85(0.36) |
| Chemical-random4 | 0.61(0.49) | 0.57(0.50) | 0.52(0.50) | 0.80(0.40) | 0.76(0.43) | 0.88(0.32) | 0.85(0.36) |
| Chemical-random5 | 0.61(0.49) | 0.55(0.50) | 0.54(0.50) | 0.79(0.41) | 0.76(0.43) | 0.92(0.27) | 0.84(0.36) |
| Chemical-random6 | 0.59(0.49) | 0.57(0.50) | 0.59(0.49) | 0.81(0.39) | 0.78(0.42) | 0.89(0.31) | 0.83(0.38) |
| Chemical-random7 | 0.61(0.49) | 0.58(0.49) | 0.54(0.50) | 0.80(0.40) | 0.81(0.40) | 0.90(0.30) | 0.82(0.39) |
| Chemical-random8 | 0.60(0.49) | 0.57(0.50) | 0.54(0.50) | 0.80(0.40) | 0.75(0.43) | 0.88(0.32) | 0.84(0.37) |

Table 11: ood noise prob0.3 mpe for each env

| Environment | Modular | GNN | RNN | FCDL | kmeans+CDL | Oracle | SAM |
|---|---|---|---|---|---|---|---|
| Chemical-random0 | 0.34(0.48) | 0.32(0.47) | 0.34(0.47) | 0.61(0.49) | 0.63(0.49) | 0.78(0.42) | 0.72(0.45) |
| Chemical-random1 | 0.36(0.48) | 0.34(0.48) | 0.33(0.47) | 0.63(0.48) | 0.64(0.48) | 0.78(0.42) | 0.76(0.43) |
| Chemical-random2 | 0.36(0.48) | 0.35(0.48) | 0.32(0.47) | 0.60(0.49) | 0.68(0.47) | 0.87(0.34) | 0.70(0.46) |
| Chemical-random3 | 0.38(0.49) | 0.36(0.48) | 0.33(0.47) | 0.60(0.49) | 0.64(0.48) | 0.90(0.30) | 0.75(0.44) |
| Chemical-random4 | 0.35(0.48) | 0.33(0.47) | 0.31(0.46) | 0.62(0.49) | 0.65(0.48) | 0.78(0.41) | 0.75(0.43) |
| Chemical-random5 | 0.37(0.48) | 0.32(0.47) | 0.32(0.47) | 0.60(0.49) | 0.63(0.48) | 0.88(0.32) | 0.76(0.43) |
| Chemical-random6 | 0.35(0.48) | 0.33(0.47) | 0.35(0.48) | 0.64(0.48) | 0.65(0.48) | 0.81(0.39) | 0.72(0.45) |
| Chemical-random7 | 0.38(0.49) | 0.34(0.48) | 0.31(0.47) | 0.63(0.48) | 0.69(0.46) | 0.81(0.40) | 0.71(0.46) |
| Chemical-random8 | 0.36(0.48) | 0.31(0.47) | 0.33(0.47) | 0.62(0.49) | 0.63(0.48) | 0.76(0.43) | 0.73(0.44) |

Table 12: ood noise prob0.6 mpe for each env

| Environment | Modular | GNN | RNN | FCDL | kmeans+CDL | Oracle | SAM |
|---|---|---|---|---|---|---|---|
| Chemical-random0 | 0.30(0.46) | 0.28(0.45) | 0.31(0.46) | 0.48(0.50) | 0.54(0.50) | 0.76(0.43) | 0.70(0.46) |
| Chemical-random1 | 0.31(0.47) | 0.31(0.46) | 0.29(0.46) | 0.50(0.50) | 0.56(0.50) | 0.73(0.45) | 0.73(0.45) |
| Chemical-random2 | 0.31(0.47) | 0.31(0.47) | 0.29(0.45) | 0.46(0.50) | 0.59(0.49) | 0.82(0.39) | 0.66(0.48) |
| Chemical-random3 | 0.33(0.47) | 0.33(0.47) | 0.30(0.46) | 0.46(0.50) | 0.55(0.50) | 0.88(0.32) | 0.74(0.44) |
| Chemical-random4 | 0.30(0.46) | 0.30(0.46) | 0.28(0.45) | 0.48(0.50) | 0.56(0.50) | 0.75(0.44) | 0.73(0.45) |
| Chemical-random5 | 0.32(0.47) | 0.30(0.46) | 0.29(0.46) | 0.47(0.50) | 0.55(0.50) | 0.85(0.36) | 0.75(0.43) |
| Chemical-random6 | 0.32(0.47) | 0.29(0.46) | 0.31(0.46) | 0.53(0.50) | 0.57(0.50) | 0.75(0.43) | 0.69(0.46) |
| Chemical-random7 | 0.34(0.48) | 0.31(0.46) | 0.29(0.45) | 0.52(0.50) | 0.61(0.49) | 0.78(0.42) | 0.67(0.47) |
| Chemical-random8 | 0.31(0.47) | 0.27(0.45) | 0.30(0.46) | 0.49(0.50) | 0.54(0.50) | 0.70(0.46) | 0.69(0.46) |

actions and the position of the obstacle. Under different conditions, the same obstacle may exhibit variations in mobility and obstructive properties, resulting in different causal graphs.

### C.2 GROUND TRUTH CAUSAL MASK

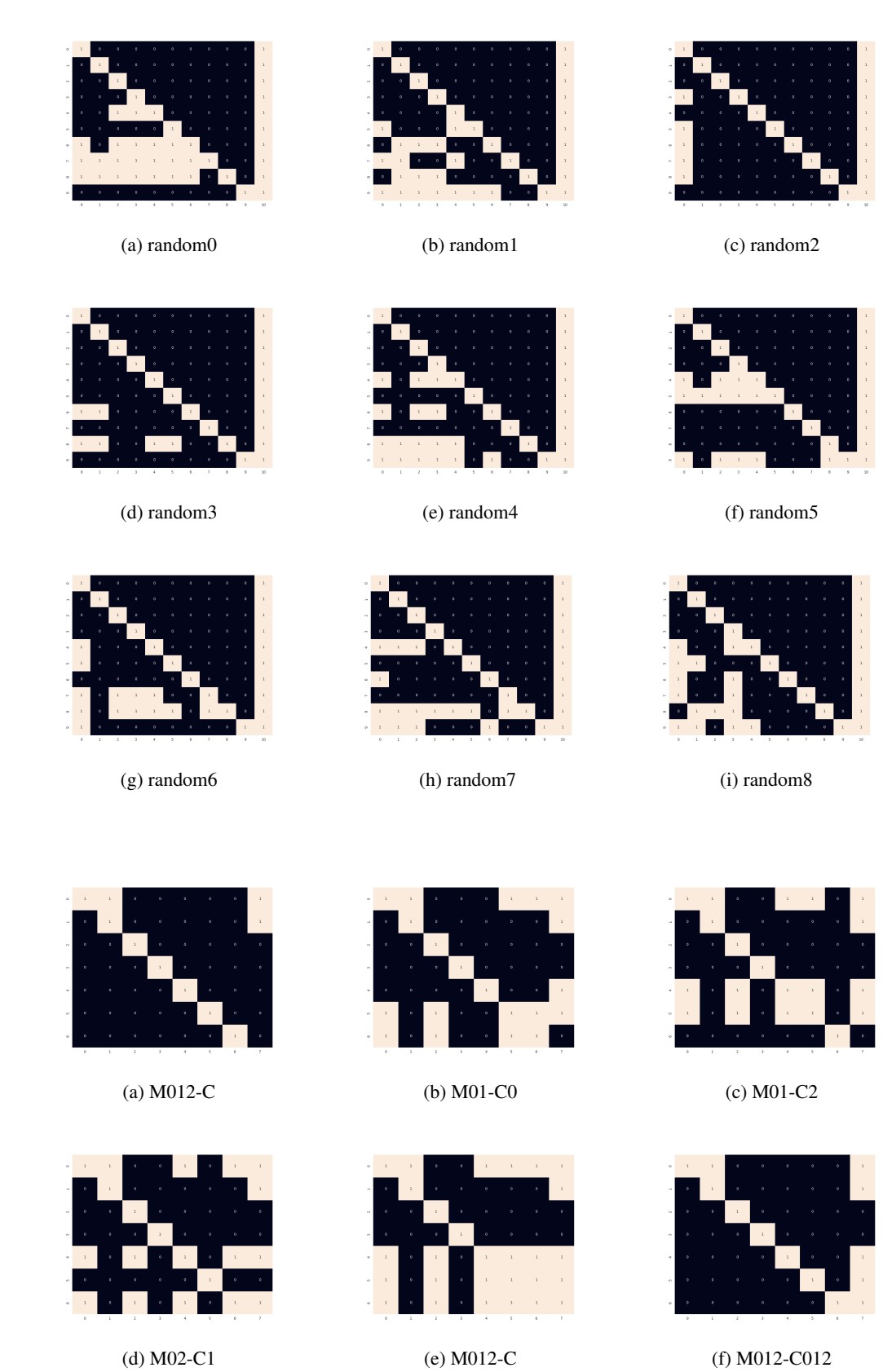

(a) random0     (b) random1     (c) random2

(d) random3     (e) random4     (f) random5

(g) random6     (h) random7     (i) random8

(a) M012-C     (b) M01-C0     (c) M01-C2

(d) M02-C1     (e) M012-C     (f) M012-C012

