# OpenReview forum: "Learning Generalizable Environment Models via Discovering Superposed Causal Relationships"
_ICLR.cc/2025/Conference — Submitted to ICLR 2025_

### Official Review · Reviewer_btSE · 2024-11-01

**Soundness:** 3
**Presentation:** 2
**Contribution:** 2
**Rating:** 3
**Confidence:** 4

**Summary:**

The paper proposes "Superposed cAusal Model" (SAM) learning. It is a framework that learns a transformer-based model which can recognize the causal relationships that the agent is encountering. It is based on an existing work and tackles the setting where trajectories are collected from different environments.

**Strengths:**

The paper tackles an interesting problem.

**Weaknesses:**

- The writing is not strong at a point that prevents getting easily the information out of the paper. In the abstract alone there are two errors: (1) a sentence with both "in this paper" and "in this article" and (2) "identify' in the sentence "where SAM shows powerful identify abilities in environments with superposed causal relationships". Errors also happens at other places in the text: "In this section, we *introduce propose* Superposed cAusal Model (SAM) learning" in the beginning of section 4 or line 214 "we design *an* simple yet efficient". In Figure 1 "Super-post" instead of superposed?
- Some words are not clearly defined such as the word "decomposition" in line 182 "This superposed causal dataset is collected from C environments that share the same decomposition but exhibit different causal relationships $\mathcal G_i$".
- The formalization is somewhat unclear at places. For instance, the learnable parameters are $\phi$, $\theta_1$ and $\theta_2$ but only $\phi$ and $\theta$ appear in Equations 1 and 2. The reader has to guess that $\theta$ is $\\{\theta_1, \theta_2\\}$. In the formalization $\mathcal S$ represents the state space of the MDP but
- The key contributions are not fully clear, which might be due to the fact that it is not clearly highlighted, particularly in the methodology section. In that section, it is written "We derive the optimization objective of the superposed causal world model using a similar approach to Varambally et al. (2024)" and then follows two paragraphs in the methodology and it is unclear what the key differences are.

**Questions:**

- What is the specific architecture of the different NN/transformers components? I do not find them in the paper nor in the appendix.

---

> ### Author Response · Authors · 2024-12-02
>
> Thank you for your evaluation of our work and for your valuable suggestions. We have incorporated your feedback into the revised paper, which is available via the link in the General Response.
>
> **W1:** The writing is not strong at a point that prevents getting easily the information out of the paper.
>
> **A:** We appreciate your insightful comments and thank you for highlighting these important issues. In response, we have carefully addressed each point and made revisions in line with your suggestions. Specifically, we have refined several expressions throughout the manuscript to improve clarity and precision. Additionally, we have reorganized Section 3 (Preliminaries) and Section 4.1 (Problem Description) to enhance the logical flow and readability. Furthermore, we have included two new diagrams: a Data Generation Process diagram (Figure 2) and a Network Architecture diagram (Figure 3), to provide clearer visual representations of the process and model structure.
>
> **W2:** Some words are not clearly defined.
>
> **A:** Thank you for your comment. We apologize for any lack of clarity and carefully revise our expression . In line 182, the term "decomposition" refers to the fact that the C environments share the same underlying state-action space structure, but each environment may exhibit different causal relationships. We have revised the manuscript to provide a clearer definition of "decomposition" to ensure it is better understood in this context.
>
> **W3:** The formalization is somewhat unclear at places.
>
> **A:** We have revised the formalization to improve its clarity and rigor. Specifically, we 1) standardized the presentation to ensure a more formal structure[L 143-150, 162-176, 207-215], and 2) added several diagrams(figure 2,3) to aid in the understanding of the formalization.
>
> **W4:** The key contributions are not fully clear
>
> **A:** Thank you for sharing your valuable perspectives. This is the first study to incorporate superposed causal relations in decision-making, where multiple causal relationships are considered for a single state-action pair. Noting that while the significance of this topic has been recognized in the field of causality (e.g., Varambally et al., 2024), studies in reinforcement learning (RL) remain limited. Some existing works in RL, particularly those on local causal graphs (LCGs), are related but differ fundamentally from our approach. For instance, a local causal model aims to learn a mask function $M: S \times A \to \{L_i\}$, where $ L_i \subset S \times A$, that maps each state-action pair $(s, a)$ to the adjacency matrix of $\mathcal{G}_L$ (Pitis et al., 2022).This assumes a unique causal graph for each state-action pair, whereas our approach considers multiple causal relationships simultaneously. We have also expanded the discussion of related work and highlighted how our method contrasts with existing approaches, as detailed in [L103-107]. Notably, our method is distinguished by its simplicity and efficiency.
>
> **Q1:** the specific architecture of the different NN/transformers components.
>
> **A:**  We are grateful for your constructive feedback. Please refer to Figure 3 for the updated diagram.  The causal graph section illustrates how trajectory data is utilized to predict the causal graph, while the dynamics model section highlights the components of the network responsible for generating predictions for the next time step based on the causal graph and the data at the current time step. We hope it will contribute to a more thorough and nuanced understanding of our method.

---

### Official Review · Reviewer_zFEd · 2024-11-02

**Soundness:** 3
**Presentation:** 2
**Contribution:** 1
**Rating:** 3
**Confidence:** 4

**Summary:**

The authors tackle the problem of learning causal structure from observational data, in an environment composed of mixtures of causal graphs. The setting is challenging and relevant, since the relationship between causes and effects in the real world can change over time, or change based on which states the trajectory passes through. The authors propose a transformer-based method that conditions on an observed trajectory to predict edge in the next-step transition dynamics, then models the next-step state accordingly. Empirical proof of concept is provided on two datasets where discerning between multiple possible causal graphs is needed for success.

**Strengths:**

The setting is ambitious and meaningful progress along this direction could be impactful in the long run

The authors show how prior work on static causal discovery can be extended to mixtures over causal graphs

**Weaknesses:**

The authors place a lot of emphasis on the novelty of their approach
[L050-051, L110-111, L523-524], but I do not agree that the idea of modeling mixtures of causal graphs is completely new. Relevant prior work is not discussed.

Clarity of presentation when it comes to the formal framework is lacking. The authors take a variational inference approach to inferring the mixtures components and local causal structure, but do not precisely define their assumptions about how the data are sampled.

**Questions:**

Comments/questions:
* [L 024, "powerful identify abilities"] typo
* [L050-051, "identifying mixtures of causal graphs has not been extensively explored", L110-111, "we are the first to address superimposed causal relationships", L523-524 "SAM addresses the limitations of existing causal world models, which assume a single causal structure governs the entire dataset"] These claims are misleading in my opinion. The framework of local causal models introduced in the CoDA paper seems quite related [https://proceedings.neurips.cc/paper/2020/hash/294e09f267683c7ddc6cc5134a7e68a8-Abstract.html]. This paper also discusses how transformers and dynamics models with sparsity penalties can be used to attempt to infer local causal structure.  work on learning from block MDPs is also relevant and should be cited [https://openreview.net/forum?id=fmOOI2a3tQP].  Bayesian inference over mixtures of graphs has also been attempted [https://proceedings.mlr.press/v180/deleu22a.html].
*[L123-130, L130-135] redundant information
* [L176-188] The formal presentation of the framework could be more precise. The authors describe how the masks are constructed in plain language but I found it difficult to ground the idea in the notation. Can an expression for masks be provided? The transition dynamics include a product over t factors, but it is not clear how the masks fit into this product. Strictly based on the writing it also seems ambiguous whether the environment is described by one mask [L174] or several [L176]. From the inference method described in L201 and L211 it seems clear that the masks are implicitly trajectory dependent, but I am not actually seeing this dependency described explicitly in the data generative process. Is $\mathcal{G}$ a random variable? Where is the idea of a mixture of causal mechanisms [L016, L531] introduced formally?
* [L293-295] while it's true that conditioning on an entire trajectory provides more information, it also would be a more complex posterior to approximate. Have the authors tried an ablation of their method where they control for the length of trajectory used for structure inference/identification?
* It would be interesting to see how the proposed method does when the underlying causal graph is static.

---

> ### Author Response · Authors · 2024-12-02
>
> Thank you for your positive evaluation of our work and for your valuable suggestions. We have incorporated your feedback into the revised paper, which is available via the link in the General Response.
>
> **W1:**   Novelty of approach and discussion of relevant prior work
>
> **A:** We would like to emphasize that, while the significance of this topic has been recognized in the field of causality (e.g., Varambally et al., 2024), studies in reinforcement learning (RL) are relatively limited, with only a few works (specifically addressing the LCG setting). Notably, one of the baselines in our study is based on the LCG, as detailed in FCDL. We have also supplemented a discussion on the related work [L102-107]
>
>  Local causal models are indeed related, but they differ from superposed causal relations. A local causal model aims to learn a mask function $M: S \times A \to \{L_i\}$, where $ L_i \subset S \times A$, that maps each state-action pair $(s, a)$ to the adjacency matrix of $\mathcal{G}_L$ (Pitis et al., 2022). This approach assumes that the causal graph is unique for the current state-action pair. However, in practice, states may be only partially observed, which may lead to multiple causal relationships for a single state-action pair. As a result, we aim to infer the causal graph by leveraging causal transition information from historical trajectories.
>
> **w2，Q3:** More precise formal presentation of the framework
>
> **A:** We appreciate the valuable feedback. We have supplemented the data generating process (figure 2) and the network diagram(figure 3) .
>
> An expression for the masks and  how the masks fit into this product are shown in network diagram.
>
> Regarding the point "whether the environment is described by one mask or several ." we have revised the manuscript by presenting the modeling of the single mask environment as a separate subsection within the preliminary section, and in our method section, we now only introduce the dataset collected from multiple environments thus including several masks. We believe this revision makes the explanation clearer.
>
> In response to the query on ”the relation between masks and trajectories” as well as “is $\\mathcal{G}\$ a random variable”, we have added a diagram depicting the data generation and inference process (see figure2). The masks are trajectory-independent, and we use the trajectory to infer the mask.
>
>  Learning a mixture of causal mechanisms refers to the process of learning from a dataset that contains multiple causal relationships[L167-169].
>
> **Q2:** Misleading claims and requiring more related work
>
> **A:** Thank you for your insightful comments. We have revised the misleading claim and included the relevant related work as suggested.
>
> Regarding LCG, we have added further details in [L103-107].
>
> For the Bayesian approach, additional information has been included in [L113-114 ].
>
> Regarding block MDPs, we have clarified that a substantial number of the related works we discuss are implemented under the factored MDP framework, as outlined in [L097-102].
>
> **Q4:**  control for the length of trajectory used for structure inference/identification
>
> **A:** Thank you for your valuable input. This is an interesting question. We will address it in future discussions.

---

### Official Review · Reviewer_Ynfc · 2024-11-04

**Soundness:** 3
**Presentation:** 2
**Contribution:** 2
**Rating:** 3
**Confidence:** 4

**Summary:**

This paper addresses the challenge of generalizing reinforcement learning models in environments with superposed causal relationships, which are mixtures of different causal mechanisms. The authors highlight the limitations of global causal discovery techniques in such settings and propose Superposed cAusal Model (SAM) learning. SAM is an end-to-end framework utilizing a transformer-based model to dynamically recognize and adapt to encountered causal relationships. Experiments in two simulated environments demonstrate SAM's effective identification abilities and generalization to unseen states.

**Strengths:**

(1) The authors introduce an algorithm, i.e., SAM, that dynamically identifies causal relationships within each episode, enabling more accurate predictions and transitions.

(2) By leveraging the Transformer architecture, the SAM can infer causal relationships from past interaction trajectories.

(3) The effectiveness is validated by two simulated envoriments.

**Weaknesses:**

(1) The paper lacks clarity in distinguishing between causal and predictive aspects of the model. Further elaboration is needed on how the proposed method identifies causal relationships and how the accuracy of these causal inferences is verified. More rigorous testing and validation are required to demonstrate the correctness of the identified causal factors.

(2) The experimental results presented in the paper are not sufficient enough to fully support the claims made. To strengthen the paper's findings, it is recommended to repeat the experiments for multiple times and report the average results across multiple trials. This will provide a more reliable and consistent basis for evaluating the model's performance.

(3) The paper should more clearly summarize the contributions of the proposed method compared to existing approaches. Highlighting the distinct advantages and novel aspects of SAM learning will help to better position the paper within the existing research landscape.

**Questions:**

None.

**Details Of Ethics Concerns:**

None.

---

> ### Author Response · Authors · 2024-12-02
>
> Thank you for your positive evaluation of our work and for your valuable suggestions. We have incorporated your feedback into the revised paper, which is available via the link in the General Response.
>
> **W1.1 :**  distinguishing between causal and predictive aspects of the model.
>
> **A:**  We appreciate the insightful feedback. In response, we have enhanced the network architecture diagram to clearly distinguish the causal and predictive components of the model. Please refer to Figure 3 for the updated diagram. Specifically, we have marked the sections corresponding to causal graph prediction and dynamics model prediction to provide a clearer visual distinction. The causal graph section illustrates how trajectory data is utilized to predict the causal graph, while the dynamics model section highlights the components of the network responsible for generating predictions for the next time step based on the causal graph and the data at the current time step. We aim for this differentiation to contribute to a more thorough and nuanced understanding of our method.
>
> **W1.2:**  Further clarification and rigorous testing are needed to explain how the method identifies and validates causal relationships.
>
> **A:**  Thank you for your valuable feedback. We have included a detailed explanation of the Structural Hamming Distance (SHD), which is used to evaluate the performance of our causal discovery methods. The SHD is a widely recognized metric in the causal inference community, quantifying the dissimilarity between the estimated causal graph and the true underlying causal structure.
>
> **W2 :**  repeat the experiments
>
> **A:**  We appreciate the valuable feedback. The model was tested across multiple seeds, but trained using only a single seed. Due to time constraints, we will improve this approach in future work.
>
> **W3 :** clearly summarize the contributions of the proposed method compared to existing approaches：
>
> **A:**  We have included additional related work [L103-107][L113-114].
>
> Noting that while the significance of this topic has been recognized in the field of causality (e.g., Varambally et al., 2024), studies in reinforcement learning (RL) are relatively scarce, with only a few notable papers. In the RL domain, although there are some existing multi-graph works under the setting of local causal graph, our work focus on a different situation which is not investigate. A local causal model aims to learn a mask function $M: S \times A \to \{L_i\}$, where $ L_i \subset S \times A$, that maps each state-action pair $(s, a)$ to the adjacency matrix of \mathcal ${G}_L$ (Pitis et al., 2022). This approach assumes that the causal graph is unique for the current state-action pair. However, in practice, states may be only partially observed, which may lead to multiple causal relationships for a single state-action pair that previous method can’t handle. We aim to infer the causal graph in this situation by leveraging causal transition information from historical trajectories.

---

### Author Response · Authors · 2024-12-02

We would like to sincerely thank the Area Chairs and Reviewers for their valuable time and insightful feedback. Below is a brief summary of the reviews and our responses for your convenience. The revised manuscript can be accessed via the link provided here: https://anonymous.4open.science/r/revision-pdf-FD28/Estimating_superposed_causal_relationships_for_offline_dynamics_model_learning-rebuttal.pdf

***Reviewer Acknowledgments***:

(1) The authors present SAM, an algorithm that dynamically detects causal relationships in each episode, improving prediction and transition accuracy. Using the Transformer architecture, SAM infers these relationships from past interactions.

(2) The setting is ambitious and meaningful progress in this direction could be impactful in the long run.

***Concerns  and Revision Overview*** :

(1) More clear Formalization: Some aspects of the formal framework, including the assumptions and definitions, are not fully clear, which could lead to some confusion.

- We have reorganized the preliminaries in section 3 and the problem description in section 4.1.
- We have added a data generation process diagram (Figure 2) and a network architecture diagram(Figure 3).

(2) More Comparisons about related work: The paper could better highlight how its approach compares to existing methods.

- We have revised the wording in several places throughout the paper and added a discussion of related work in Section 2.

We appreciate the constructive feedback provided.

---

### Meta-Review · Area_Chair_Dmsm · 2024-12-17

**Metareview:**

The paper focuses on the problem of generalizing RL in scenarios with superposed causal relationships. The two big issues raised by the reviewers are clarity of the exposition as well as limitations in experimental evaluation.

My decision is based on the assessment that the exposition can be improved significantly. Also all the reviewers rate the papers similarly without anyone willing to argue for the paper.

**Additional Comments On Reviewer Discussion:**

All the reviewers agree on rejecting the submission – while the authors posted rebuttals, none of the reviewers engaged in further discussion. I took sometime to look at the details and I do agree that the paper could be presented in a clearer manner, specifically highlighting and focusing on the novelty. Not having any reviewer as a champion further makes it hard for me to consider anything else except a rejection.

---

### Decision · Program_Chairs · 2025-01-22

Reject